# Effect of sodium hydroxide dosage on strength development in cement-fly ash mortars: Experimental and ANN-based prediction

Jing Xu[1,2☯], Fei Peng Liu[1,3,4☯]*, Jian Xin Zhao[3,4], Shu Cheng Tan[1,5,6]*, Ai Min Gong[7]

1 Institute of International Rivers and Eco-Security, Yunnan University, Kunming, China, 2 Great Lakes Institute for Environmental Research (GLIER), University of Windsor, Windsor, Canada, 3 Southwest Investigation and Planning Institute, National Forestry and Grassland Administration, Kunming, China, 4 National Forestry and Grassland Administration's Scientific Research and Monitoring Base for Forests and Grasslands on the Qinghai–Tibet Plateau, Beijing, China, 5 School of Earth Science, Yunnan University, Kunming, China, 6 Yunnan International Joint Laboratory of Critical Mineral Resource, Kunming, China, 7 College of Water Conservancy, Yunnan Agricultural University, Kunming, China

☯ These authors contributed equally to this work.
* lfp881214@126.com (FPL); shchtan@ynu.edu.cn (SCT)

## Abstract

Fly ash, a by product from coal and biomass combustion in fossil-fuel power plants, is composed of fine particulate matter that can contribute to terrestrial and atmospheric pollution if not properly contained. In this study, we explored the use of an alkali activator in the production of fly ash-based concrete to contain fly ash particles. Fly ash was mixed with different concentrations of an alkaline activator (sodium hydroxide) for up to 28 days. We employed SEM images and laboratory tests supported by an Artificial Neural Network (ANN) to determine the properties and strength of concrete. Our findings reveal that sodium hydroxide at 5 and 6% optimized the strength of colloidal sand mixture and the precursor of concrete at 3 and 28 days, respectively. Fly ash content of 10% optimized concrete strength, while 20% and 30% fly ash contents at 3 and 28 days resulted in better cement strength compared to the control. Sodium hydroxide initially rapidly improved colloidal sand mixture strength but its influence tapered over time. Amorphous silica and alumina phases significantly affected the performance of ordinary fly ash alkali-activated mortar. ANN learning training data effectively assisted laboratory tests to determine the strength of concrete with a percent error of 10, demonstrating the potential of this approach in enhancing the understanding of concrete properties and strength development.

## 1. Introduction

Ordinary Portland cement is an important material in the construction industry. The production of Ordinary Portland cement (OPC) in China has soared from 2.9 billion tonnes in 2008 to 4.65 billion tonnes in 2016 [1]. However, the production of Ordinary Portland cement is one of the major contributors to greenhouse gas emissions,

**Data availability statement:** The datasets generated and analyzed during this study contain

potentially identifiable or sensitive personal information and are therefore not publicly available. Qualified researchers may request access by contacting the Institute of International Rivers and Eco-Security, Yunnan University (Tel: +86-871-65034577; +86-871-65940990) or the corresponding author (lfp881214@126.com; shchtan@ynu.edu.cn).

**Funding:** This research was supported by China Canada Joint Water Ecology and Watershed Management Innovation Talent Training Project (grant NO. CXXM20190105), the Famous Teacher of Xing Dian Talent Teacher's Program of Yunnan Province Education Department (Grant NO. XDTT202206), Yunnan Fundamental Research Projects (Grant NO. 202301BF070001-020), the Yunnan Province Science and Technology Department (No. 202101BA070001-145), the Yunnan Province Science and Technology Department (No. 202101BA070001-145), Yunnan Key research and development plan program (Grant NO. 202203AP140077), the Science and Technology Innovation Team Program of Yunnan Province Education Department (Grant NO. CY22624109), and the Graduate Tutor Team Program of Yunnan Province Education Department (Grant NO. CY22622205) . The funders provided financial support but had no role in study design, data collection and analysis, decision to publish, or preparation of the manuscript.

**Competing interests:** No authors have competing interests.

**Abbreviations:** AAM, Alkali–Activated Materials; BFA, Biomass Fly Ash (derived from combustion of agricultural biomass); C-(A)-S-H, calcium-(alumino)-silicate-hydrate; CC, Calcium Carbonate($CaCO_3$); CFA, Co-combustion Fly Ash (derived from combustion of wooden biomass and coal); CH, Calcium Hydroxide ($Ca(OH)_2$; C10, C15, 10% or 15% substitution by mass of fly ash with quicklime; EDS, Energy Dispersive X-Ray Spectroscopy; FA, Fly Ash; FE, SEM-Field Emission Scanning Electron Microscope; H, Bound $H_2O$; LOI, Lost on Ignition measurement; N-A-S-H, sodium-alumino-silicate-hydrate; N5, N10, 5M or 10M solution of NaOH; OPC, Ordinary Portland Cement; PSDs, particle size distributions; SSA, specific surface area; S22, 100 g addition of sodium silicate to alkaline activator; TGA, thermogravimetric analysis; XRD, X-ray diffraction test; XRF, X-ray fluorescence test; A1B5, "X" is FA number, "Y" is the symbol for the NaOH number (X=1 for the 10% FA and Y=4 for the 5%NaOH in Table 2).

accounting for 6–8 percent of all man-made carbon dioxide emissions [2,3]. The increase in the carbon footprint is mainly related to its industry's high fuel demand (largely met by fossil fuels) and chemical production processes, which release large amounts of carbon dioxide as a by-product [4].

The growing global awareness of greenhouse gas emissions from energy and industrial processes has prompted the adoption of various mitigation strategies [5], including increasing the share of renewable energy in energy production, use of biomass fuels, improvements in equipment efficiency, preparation of eco-friendly concrete components, utilization of alternative raw materials, and material substitution [6,7].

One way to reduce $CO_2$ emissions from Ordinary Portland cement production is to develop a new type of sustainable alternative material derived from industrial waste, such as fly ash [7]. Fly ash composites may be more environmentally friendly than Ordinary Portland cement because the former is sourced mainly from industrial waste, require less energy to produce, and cause less carbon dioxide [8]. These fly ash composites are comparable to OPC materials because they have excellent chemical resistance, adequate compressive strength, and slight drying shrinkage [9]. To obtain fly ash composites with comprehensive properties, the choice of suitable raw material combined with an alkaline solution of proper molar concentration and Si/Al ratio, and curation at either room or high temperature are critical. Metakaolin, fly ash, or slag are the most commonly utilized as binders.

The rapid development of the power industry increased the emissions of fly ash. In China, this pattern is evident in the total amount of fly ash emitted by coal-powered plants over 15 years: about 125 million tons of coal ash were emitted in 1995, and 300 in 2010 [10]. Fly ash contains harmful substances (natural radionuclides such as Ra), which pollute water, soil, and air [11,12]. Therefore, the increasing use of fly ash waste could reduce its concentration in atmospheric and terrestrial ecosystems.

Properties, especially material strength, are essential to the durability and quality of concrete structures [13]. The development of concrete strength is the result of hydration [14,15]. Factors affecting the formation of C-S-H can influence the strength of concrete. Fly ash is capable of increasing the formation of C-S-H [16] and a high $SiO_2$ and $Al_2O_3$ content of fly ash results from the pozzolanic reaction (Table 1). The vitreous body is the main source of fly ash "potential" activity [17], and the slow reaction of pozzolanic leads to its reduction in early strength and increase in the later strength of concrete [12,18].

Fly ash requires alkaline activators, of which sodium silicate and sodium hydroxide (NaOH) are most commonly used. This kind of activator used has a great influence on the dissolution of silicon and aluminum ions. Appropriate amounts of alkali activate the early activity, enhance the compressive strength of concrete at both early and late stages, and improve its impermeability, durability, and corrosion resistance [19]. These reactions are caused by the breaking apart of Al-O and Si-O bonds in the slag by an alkali activator, accelerating the dissolution of aluminum and silicon ions [20]. Jaarsveld (2002) [21] pointed out that using a sodium hydroxide solution in the fly ash bonding process leads to higher strength and coagulation and lower acid

**Table 1. Physical performance of fly ash (Ⅱ level) (%).**

| Fineness | Water demand ratio | Loss on ignition | Water | SO₃ |
|---|---|---|---|---|
| 13.1 | 104 | 2.2 | 0.1 | 0.73 |

resistance than that obtained by using a potassium hydroxide solution. The use of NaOH and sodium silicate can increase the silicon content of the concrete mixture, thereby improving the mechanical properties of concrete. The most commonly used molar concentration of sodium hydroxide in experimental conditions should not be more than 20M, and sodium silicate solution to NaOH content should not exceed 1.0 [6].

In fly ash-based geopolymers, the analysis of the mechanism of geopolymerization is complicated due to the existence of crystalline phases (hematite, mullite or quartz). According to Fernandez - Jimenez (2006) [22], fly ash composition is an important factor in the rational geopolymer binding process. The lowest levels of active silicon dioxide and alumina in fly ash compounds ensure that the binding reaction occurs.

The use of the acronym "Alkali activated materials (AAM)" is recommended instead of geopolymers for fly ash characterized by high levels of calcium compounds [23]. Geopolymers typically necessitate activation and solidification at high temperatures, whereas AAM can be used under normal environmental conditions. The main hydrate in alkali-activated materials is calcium silicate hydrate (C-(A)-S-H) in the geopolymer. C-(A)-S-H appears with sodium-alumino-silicate-hydrate (N-A-S-H) [8]. Each year, more fossil-fuel power plants are turning to co-combustion or new biomass sources. Analysis and understanding of the suitability and quality of such variable fly ash during combustion are critical for its potential use as a geopolymer binder or AAM.

Numerous studies investigated the strength properties of fly ash concrete. Zhang et al. (2021) [24] found that fly ash improves cement strength. Many factors can affect concrete quality, including water-cement ratio, curing age, and proportion of coarse and fine aggregates, etc. However, a scientific system is needed to predict their impacts on concrete. To do that, a data mining model acts as a suitable tool to predict the performance of concrete for different compositions of raw materials.

As data mining models, Artificial Neural Networks (ANNs) have been widely used in civil engineering, especially in building materials [2,17,25,26]. An ANN has an excellent ability in approximate function and pattern recognition, which are the typical characteristics of concrete materials. In addition, advance knowledge of the mix or properties of concrete is not needed, since the exact links between output and input parameters are unknown, and they can be identified and the connections can be learned through model training [27,28].

Compressive strength is the most important and basic property of concrete, and it can affect other properties of concrete [29]. Huang (2022) [30] used cement, fly ash, and a water reducer as factors to affect the compressive strength of concrete. Khademi (2016) [31] used a neural network to rank seven compressive strength factors of recycled concrete. In addition, Ahmed et al. (2023) [32] used an ANN to predict the compressive strength of concrete modified with nanoparticles. On the other hand, Ahmed et al. (2017) [33] used an ANN to predict the width of cracks in concrete. These earlier models were used in this study as a theoretical basis of concrete mix proportion and engineering application, and to build the neural network.

ANNs have strong information processing ability and knowledge because the goal behind them is to reflect the thinking process of the human brain [34]. They can learn and generalize from training parameters, and have the ability to learn the mathematical relationships between the output and input parameters. The first ANN was called multilayer feedforward network [35]. Other networks include self-organizing feature map [36], hopfield network [37], and radial basis function networks, etc.

To assess whether biomass and co-fired ash can be used as an alkali-activated material, a sodium hydroxide activator was used in activation of the analyzed fly ash. The slurry and mortar were analyzed using a scanning electron microscope (SEM). We also explored the application of a neural network in predicting concrete performance, various input

factors, with strength taken as the sole output parameter. Understanding the effects of alkali activators on fly ash from coal combustion is important for analyzing its potential reuse as AAM precursors and its possible application as engineering material.

## 2. Materials and methods

### 2.1. Materials

Cement: Ordinary Portland Cement (OPC)

Ordinary Portland cement used in this study was produced by Yunnan Dongjun cement Co., Ltd. All indexes met those of the General Portland cement standards (GB 175) [38];

Fly-ash was produced by Yunnan Hengyang Industrial Co., Ltd, and its performance indexes are shown in Table 1;

Fine aggregate was standard sand;

The water used in the experiment is municipal tap water (pH = 7.2 ± 0.3, conductivity < 100 μS/cm);

Alkali used was sodium hydroxide produced by Tianjin FengChuan chemical reagent Technology Co., Ltd., with NaOH content is no less than 96.0% and carbonate ≤1.5% (When dissolved in water, carbonate and sodium hydroxide disassociate into carbonate and metal cations, and sodium and hydroxide ions respectively, which move freely to form a homogeneous solution).

### 2.2. Methods

**Density.** True density of fly ash was measured in accordance with GB/T 1596 [39] using a gas pycnometer (produced in Beijing). The device operates by detecting pressure changes caused by gas displacement of a solid object moving between a sample chamber and a reference chamber.

Before preparing the mortar, the alkali activator was dissolved in water and allowed to cool to room temperature (Based on my preliminary experiments, the sodium hydroxide concentration of 3% to 8% was selected). Mortar was made in specific proportions and vibrated. All alkali activated mortar mixtures were prepared and manufactured according to the modified GBT-17671 [40] procedure as follows:

The aluminum silicate precursor was added to the mixing plate and the agitator was started at a low rotational speed (rate −120 ± 5 rpm-1), Sand was added at a uniform speed in the first 45 seconds of mixing, The basic activator solution was added at a uniform rate (time measurement was set to "Zero Time") in the next 45 seconds of mixing, Mixer speed was set to high speed (rotational motion 300 ± 10 rpm) and mixing continued for more than 45 seconds, Mixing stopped after a total of 135 seconds following GBT-17671 [40] standard mortar preparation procedures.

Samples were maintained initially under laboratory conditions for 4 hours and then solidified in a 70°C dryer for 4 hours. After heat curing, all samples were maintained under laboratory conditions (temperature of 20 ± 2°C and relative humidity of 90%) until testing(According to the GB/T 17671 [40] standard, this temperature and humidity range ensures hydration stability of specimens and comparability of test results). Samples were removed from the mold after 24(±3) hours and maintained under the above laboratory conditions. After curing, specimens of their mechanical properties were tested using the Chinese ISO method (GBT17671) [40].

Fly ash was mixed with an alkaline activator according to the above procedure, using sodium hydroxide to create a hydroxide aqueous solution. Sample names (Table 2) are based on the universal symbol "10% XXX-3% y", where "XXX" is FA, "y" is the symbol for the alkali activator NaOH (3% for the amount of sodium hydroxide, 10% and C15 for the 10% substitution of FA for cement). Properties of alkali activated materials were obtained using bending and compression tests and a scanning electron microscope (SEM) [41].

**SEM analysis.** Thermal field emission scanning electron microscope (KYKY-EM3200 produced by Beijing China Science and Technology Instruments Co.) was used for morphological analysis of fly ash and mortar samples. A mortar sample was taken from 150 × 150 × 150 mm beam samples. A 1-cm thick sample was cut from the beam, dried, and

**Table 2. Content of NaOH affects the active ratio of fly ash and the flexural and compressive strength of fly ash at different ages.**

| Order number | Proportion of fly ash (F) and NaOH | W/C | Cement/g | Fly ash/g | NaOH/g | Standard sand/g | Water/ml | Flexural(3d)/MPa | Compression(3d)/MPa | Flexural(28d)/MPa | Compression(28d)/MPa |
|---|---|---|---|---|---|---|---|---|---|---|---|
| 1–1 | 10%F | 0.5 | 405 | 45 | — | 1350 | 225 | 4.5 | 20.6 | 8.9 | 47.9 |
| 1–2 | 10%F-3%NaOH | 0.5 | 405 | 45 | 1.35 | 1350 | 225 | 4.8 | 22.3 | 8.7 | 45.3 |
| 1–3 | 10%F-4%NaOH | 0.5 | 405 | 45 | 1.8 | 1350 | 225 | 5.3 | 22.8 | 8.8 | 46.5 |
| 1–4 | 10%F-5%NaOH | 0.5 | 405 | 45 | 2.25 | 1350 | 225 | 5.4 | 22.8 | 8.9 | 46.4 |
| 1–5 | 10%F-6%NaOH | 0.5 | 405 | 45 | 2.7 | 1350 | 225 | 5.4 | 22.5 | 8.9 | 47.2 |
| 1–6 | 10%F-7%NaOH | 0.5 | 405 | 45 | 3.15 | 1350 | 225 | 5.1 | 22 | 8.5 | 45.3 |
| 1–7 | 10%F-8%NaOH | 0.5 | 405 | 45 | 3.6 | 1350 | 225 | 4.9 | 21.5 | 8.3 | 43 |
| 2–1 | 20%F | 0.5 | 360 | 90 | — | 1350 | 225 | 4.2 | 16.7 | 8.4 | 43.9 |
| 2–2 | 20%F-3%NaOH | 0.5 | 360 | 90 | 2.7 | 1350 | 225 | 4.6 | 18.9 | 8.2 | 39 |
| 2–3 | 20%F-4%NaOH | 0.5 | 360 | 90 | 3.6 | 1350 | 225 | 4.9 | 20 | 8.3 | 39.8 |
| 2–4 | 20%F-5%NaOH | 0.5 | 360 | 90 | 4.5 | 1350 | 225 | 5.4 | 22.4 | 8.3 | 40.4 |
| 2–5 | 20%F-6%NaOH | 0.5 | 360 | 90 | 5.4 | 1350 | 225 | 4.9 | 19.5 | 8.8 | 42.3 |
| 2–6 | 20%F-7%NaOH | 0.5 | 360 | 90 | 6.3 | 1350 | 225 | 4.4 | 17.5 | 8.4 | 36.1 |
| 2–7 | 20%F-8%NaOH | 0.5 | 360 | 90 | 7.2 | 1350 | 225 | 4.3 | 17 | 7.9 | 32.9 |
| 3–1 | 30%F | 0.5 | 315 | 135 | — | 1350 | 225 | 4 | 16.4 | 8 | 35.1 |
| 3–2 | 30%F-3%NaOH | 0.5 | 315 | 135 | 4.05 | 1350 | 225 | 4.1 | 15.2 | 6.9 | 27.8 |
| 3–3 | 30%F-4%NaOH | 0.5 | 315 | 135 | 5.4 | 1350 | 225 | 4.2 | 15.9 | 7 | 28 |
| 3–4 | 30%F-5%NaOH | 0.5 | 315 | 135 | 6.75 | 1350 | 225 | 4.5 | 16.5 | 7.1 | 28.9 |
| 3–5 | 30%F-6%NaOH | 0.5 | 315 | 135 | 8.1 | 1350 | 225 | 4.1 | 13.5 | 8.4 | 37 |
| 3–6 | 30%F-7%NaOH | 0.5 | 315 | 135 | 9.45 | 1350 | 225 | 3.4 | 11.7 | 8.2 | 34 |
| 3–7 | 30%F-8%NaOH | 0.5 | 315 | 135 | 10.8 | 1350 | 225 | 2.9 | 9.5 | 7.5 | 28.6 |

Note: The sample names in Table 2 are based on the universal symbol AX-BY", where "X" is is FA number, "Y" is the symbol for the NaOH number (X = 1 for the 10% FA and Y = 4 for the 5%NaOH in Table 2).

the SEM was prepared by covering the sample with a thin gold film under vacuum. The images were taken at different magnification under high-pressure vacuum and 5 kv acceleration voltage.

**Compressive and flexural strength.** Compression and bending strength tests were performed according to standard GBT-17671 [40] using 150 × 150 × 150 mm standard specimens and a hydraulic press (YES-2000 made in China) equipped with necessary accessories. Mortar was tested 3 and 28 days after preparation. The test was performed at a loading rate of 0.2 MPa/min, a compressive strength sensitivity of 50 kN, a loading rate of 0.01 MPa/min, and a flexural strength sensitivity of 2kN(GB/T 17671) [40]. The final result is an average of the flexural and compressive strength of three specimens.

**Artificial neural network (ANN) prediction.** After laboratory tests, we introduced a neural network to predict strength. A neural network is a self-adaptive and self-learning algorithm that can be used to solve nonlinear problems. We established a neural network with an input vector composed of cement, fly ash, standard sand, water, and calcium oxide, and an output value of the concrete's compressive strength. We normalized the input and output data as follows:

$$x_i = \frac{x_i - x_{min}}{x_{max} - x_{min}}$$

(1)

The data was normalized and split into training (70%), validation (15%), and test (15%) datasets. The network architecture consisted of an input layer with five nodes, a hidden layer with ten nodes (selected after trials for optimal performance),

and an output layer with one node. We utilized the Levenberg-Marquardt (trainlm) training function for its fast convergence and effectiveness in handling nonlinear problems. The model was trained for up to 1000 epochs with a goal of achieving a mean squared error (MSE) below 0.00001. Model performance was assessed using root mean square error (RMSE), mean absolute error (MAE), and coefficient of determination ($R^2$) on both training and test datasets to ensure the model's ability to accurately predict concrete strength while avoiding overfitting through validation monitoring.

**Correlation analysis.** We used correlation to measure the degree of association between variable factors. The variable elements of concrete raw materials and test results are analyzed, so as to measure the correlation degree of each variable factor.

## 2.3. Experimental apparatus

The unconfined compressive strength tests were performed on a WAW-1000 machine in the laboratory of Yunnan University. The test apparatus as shown in Fig 1.

## 3. Results and discussion

Figs 3–6 show the average strength data of mortar mixtures at days 3 and 28.Mechanical properties of RFA-NaOH samples were not measurable on day 3 and 28. This may be due to insufficient alkaline activation; these mixtures peel and strength slowly increase over time. For FA mortars containing sodium hydroxide – activators, flexural and compressive strength increased between 7 and 28 days in most cases. With the increase in NaOH molar concentration, the strength of the studied mortar increased with time. Other researchers have observed a similar trend and concluded that the concentration of alkali plays a significant role in determining the mechanical properties of alkali-activated materials [42]. Elevated levels of calcium compounds can interfere with the process of geological polymerization and cause degradation of the microstructure of specimens [30,42].

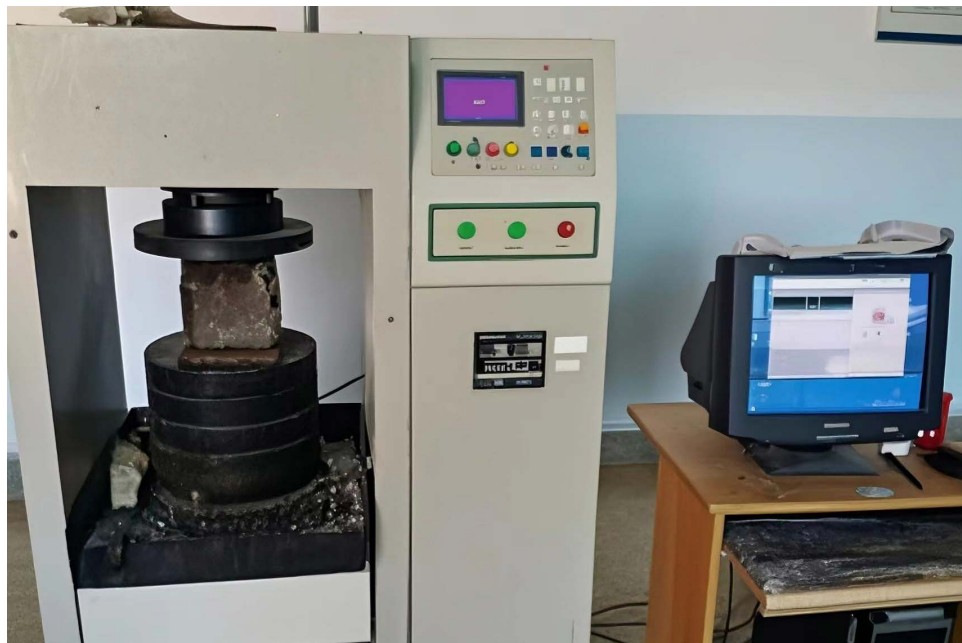

**Fig 1.  The unconfined compressive strength test device.**

## 3.1. SEM analysis of FA

A micrograph of 10 µm clearly shows the spherical shape of the full fly ash particles (Fig 2). SEM analysis confirmed that the spherical particles in RFA were primarily composed of silicon dioxide and aluminum compounds that are typical of glassy aluminosilicates. Furthermore, the ceno-spherical particles in RFA were mostly non-porous and without any defects (Fig 2).

## 3.2. Effect of sodium hydroxide content on fly ash cement strength

According to the mix proportion established by Xiaohui [17], the test block formed, and the flexural and compressive strength were measured after standard curing for 3d and 28d (Table 2). Changes in compressive strength between alkali-activated mortars (Table 2) may be affected by differences in precursor properties. For example, particle fine-ness associated with the formation of larger amounts of alkaline aluminum silicate gels affects flexural and compressive strength results [34]. For mixtures activated by a single alkali activator, the choice of raw materials seems to be more important. In a certain range, increasing NaOH concentration has a similar positive effect on RFA mortar, increasing compressive strength.

After 3 days:

The greatest strength after 3 days was obtained with 5% sodium hydroxide and 10% fly ash (Table 2).

Our results showed that the amount of alkali added in the test was not sufficient to activate the early strength of fly ash.

The optimum sodium hydroxide amount was 5% (Figs 3 and 4). With the increase in alkali content, the strength of mortar increased at first, and then decreased. Furthermore, mortar was strongest when sodium hydroxide accounted for 5% of fly ash by weight over three days, regardless of the amount of cement added.

When fly ash content in the cement was 20%, the highest compressive and flexural strength was observed at the end of 3 days. Flexural and compressive strengths increased, respectively, 20 and 11% with 10% fly ash content, 29 and 34% with 20% fly ash content, and 13 and 4% with 30% fly ash content.

The early strength of sodium hydroxide mortar developed rapidly.On the 3rd day, the strength of the alkali-added mortar was generally higher than that of non-alkali mortar (Fig 3). When the optimal combination was 1–5(number), the flexural

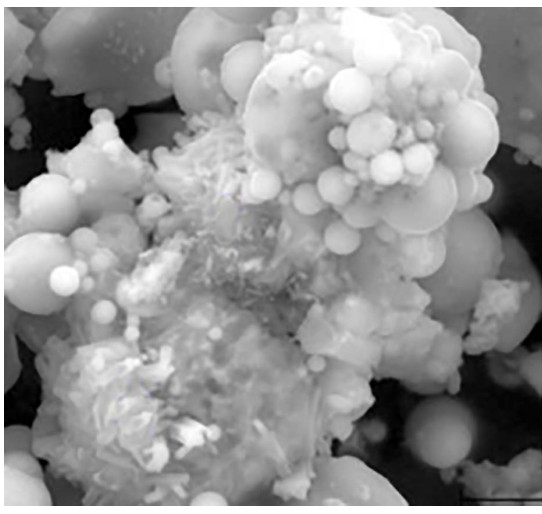

**Fig 2. SEM image of 10 µm fly ash.**

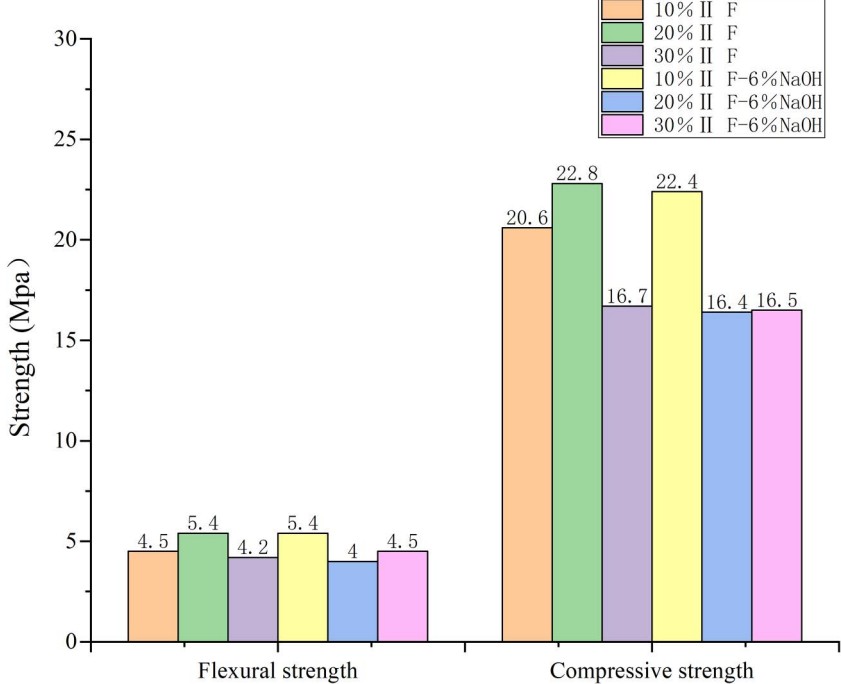

**Fig 3. Comparison of flexural and compressive strength of the tested combinations after 3 days.**

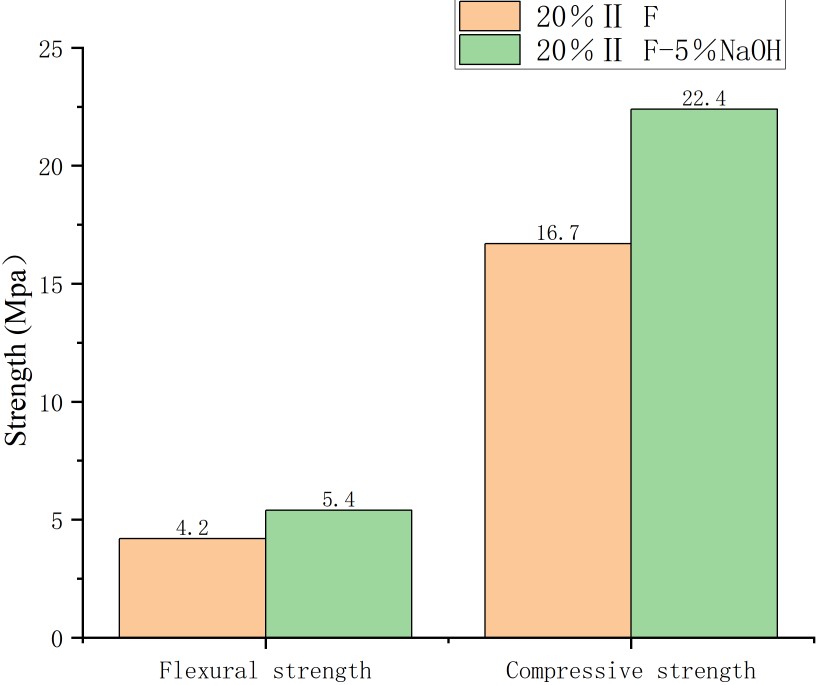

**Fig 4. 20% Ⅱ level strength of fly ash after 3 days.**

strength was 5.4 MPa compared to 4.5 MPa of the control group, and the compressive strength was 22.5 MPa, compared to 20.6 MPa of the reference group.

### 3.3. The effect of age on the strength of mortar mixed with an alkali-activator

At 28 days, the strength reached its maximum with fly ash content of 10% and NaOH content of 6% (**Table 2**).

The results (Table 2) showed that the compressive and flexural strength were highest at 3 days when the combination of fly ash content and NaOH was A1B4, and the optimal combination of fly ash content and NaOH content was A1B5 or A1B1 at 28 days; that is, 5% of sodium hydroxide added in the early stage (3 days) can effectively stimulate strength performance, while at 28 days, the strength of alkali-activated (6%) and non- activated (0%) was equivalent, indicating that the strength of the non- activated develops rapidly at a later stage. When the importance of mixing fly ash and NaOH was compared, the difference was almost double (range ratio 14.6/7.47 = 1.95) at 28 days compressive strength.

The age-related effects of sodium hydroxide were strongest in 6% of the sodium hydroxide mixtures at 28 days and in 5% of the sodium hydroxide mixtures at 3 days.

The optimum amount of alkali varied with age from 5 to 6% or 0%, as the amount of $Ca(OH)_2$ produced by early hydration was greater than that produced by later hydration.

Compared with the control group, the increase in strength was highest with 30% fly ash (after 28 days) and 20% fly ash (after 3 days).

The use of 5% NaOH and 30% fly ash resulted in the strength of fly ash mixture surpassing that of the control group after 28 days. Therefore, the optimum fly ash content was 30% at 28 days. (Fig 5).

The early strength of concrete mixed with NaOH and fly ash developed rapidly, and then slowed.

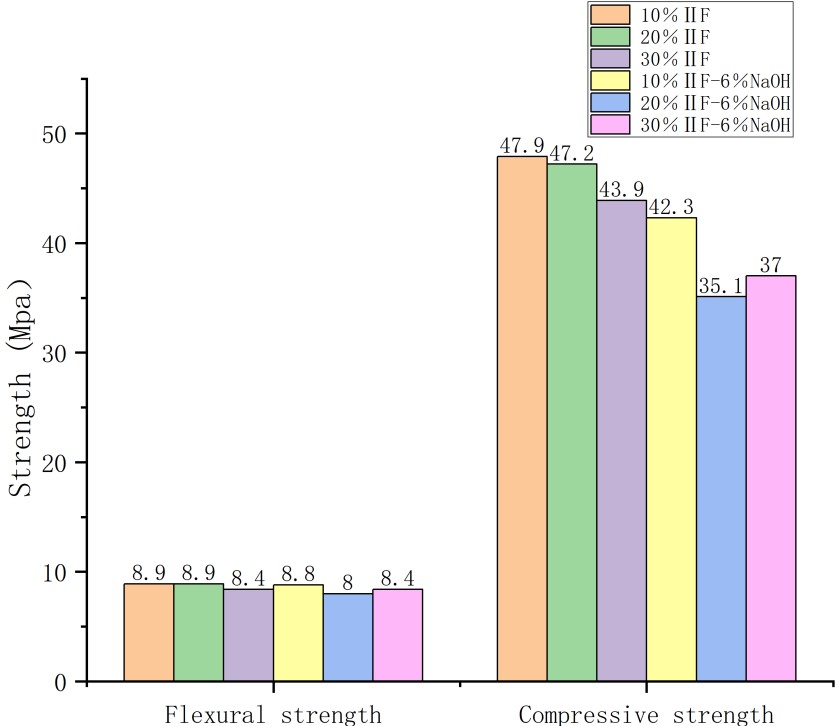

**Fig 5. A comparison of the best combinations of flexural and compressive strengths of fly ash content after 28 days.**

Under the optimal combination, the compressive and flexural strength of the mortar after 3 days was 29 and 34% higher, respectively, than that of the control group. After 28 days, the optimal combination resulted in an increase of only 5% in both compressive and flexural strengths (Figs 4 and 6).

### 3.4. Results of the neural network

We compared the predicted with the actual data to assess the performance of the prediction model (Table 3 and Fig 7). The error histogram of the neural network compressive strength prediction (Fig 7) showed the errors mainly between −0.1 and 0.1 in the first eight data groups (Table 3). The relative error of the remaining data groups was −8.4–8.3% (Table 2). To further enhance the robustness of our modeling results, we conducted a more comprehensive sensitivity analysis, examining how variations in each input variable (cement, fly ash, standard sand, water, and calcium oxide) individually and in combination affect the predicted compressive strength. This deeper exploration revealed the relative importance of each variable in the prediction process. Additionally, we incorporated statistical measures of uncertainty by calculating confidence intervals for the predictions, providing a range within which the true compressive strength is likely to fall. We also rigorously assessed the model's generalization ability by evaluating its performance on a separate test dataset that was not used during training. The results showed that the neural network effectively predicted the compressive strength of concrete with an alkali activator, even for data not encountered during training. By establishing this prediction model and incorporating uncertainty quantification, we have created a more reliable foundation for future experiments that focus on designing concrete with an alkali activator.

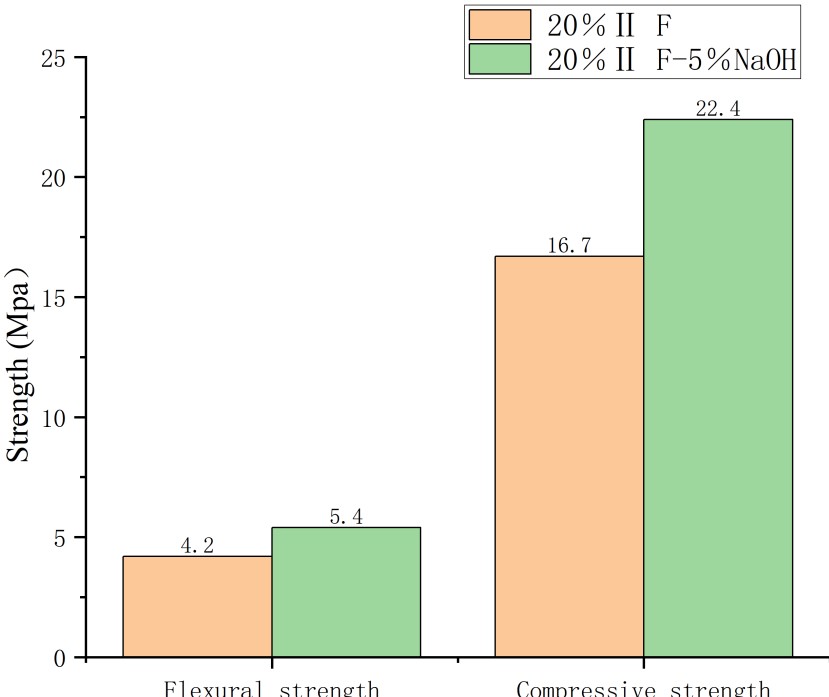

**Fig 6. The difference in 30% Ⅱ level strength of fly ash after 28 days.**

Table 3. Compressive strength data of 21 groups of different mix proportions.

| Order number | W/C | Cement (g) | Fly ash (g) | Standard sand (g) | Water (ml) | NaOH (g) | True value of compressive strength | | | | Predicted compressive strength | | | |
|---|---|---|---|---|---|---|---|---|---|---|---|---|---|---|
| | | | | | | | 3d flexural | 3d Compression | 28d flexural | 28d Compression | 3d flexural | 3d Compression | 28d flexural | 28d Compression |
| 1–1 | 0.5 | 405 | 45 | 1350 | 225 | 0.0 | 4.5 | 20.6 | 8.9 | 47.9 | 4.7 | 22.0 | 8.6 | 47.4 |
| 1–2 | 0.5 | 405 | 45 | 1350 | 225 | 1.4 | 4.8 | 22.3 | 8.7 | 45.3 | 4.8 | 22.3 | 8.7 | 46.8 |
| 1–3 | 0.5 | 405 | 45 | 1350 | 225 | 1.8 | 5.3 | 22.8 | 8.8 | 46.5 | 4.9 | 22.4 | 8.7 | 46.5 |
| 1–4 | 0.5 | 405 | 45 | 1350 | 225 | 2.3 | 5.4 | 22.8 | 8.9 | 46.4 | 5.0 | 22.4 | 8.7 | 46.1 |
| 1–5 | 0.5 | 405 | 45 | 1350 | 225 | 2.7 | 5.4 | 22.5 | 8.9 | 47.2 | 4.9 | 22.3 | 8.7 | 45.6 |
| 1–6 | 0.5 | 405 | 45 | 1350 | 225 | 3.2 | 5.1 | 22 | 8.5 | 45.3 | 4.9 | 22.2 | 8.7 | 45.1 |
| 1–7 | 0.5 | 405 | 45 | 1350 | 225 | 3.6 | 4.9 | 21.5 | 8.3 | 43 | 4.9 | 22.0 | 8.7 | 44.4 |
| 2–1 | 0.5 | 360 | 90 | 1350 | 225 | 0.0 | 4.2 | 16.7 | 8.4 | 43.9 | 4.3 | 18.1 | 7.9 | 43.1 |
| 2–2 | 0.5 | 360 | 90 | 1350 | 225 | 2.7 | 4.6 | 18.9 | 8.2 | 39 | 4.7 | 19.2 | 7.8 | 40.6 |
| 2–3 | 0.5 | 360 | 90 | 1350 | 225 | 3.6 | 4.9 | 20 | 8.3 | 39.8 | 4.7 | 19.6 | 8.0 | 41.0 |
| 2–4 | 0.5 | 360 | 90 | 1350 | 225 | 4.5 | 5.4 | 22.4 | 8.3 | 40.4 | 5.0 | 20.6 | 8.1 | 40.9 |
| 2–5 | 0.5 | 360 | 90 | 1350 | 225 | 5.4 | 4.9 | 19.5 | 8.8 | 42.3 | 4.8 | 19.1 | 8.2 | 39.6 |
| 2–6 | 0.5 | 360 | 90 | 1350 | 225 | 6.3 | 4.4 | 17.5 | 8.4 | 36.1 | 4.7 | 18.3 | 8.2 | 37.1 |
| 2–7 | 0.5 | 360 | 90 | 1350 | 225 | 7.2 | 4.3 | 17 | 7.9 | 32.9 | 4.7 | 17.6 | 8.2 | 34.0 |
| 3–1 | 0.5 | 315 | 135 | 1350 | 225 | 0.0 | 4 | 16.4 | 8 | 35.1 | 4.3 | 17.1 | 7.4 | 33.9 |
| 3–2 | 0.5 | 315 | 135 | 1350 | 225 | 4.1 | 4.1 | 15.2 | 6.9 | 27.8 | 4.3 | 14.4 | 7.0 | 28.6 |
| 3–3 | 0.5 | 315 | 135 | 1350 | 225 | 5.4 | 4.2 | 15.9 | 7 | 28 | 4.4 | 14.8 | 7.3 | 28.3 |
| 3–4 | 0.5 | 315 | 135 | 1350 | 225 | 6.8 | 4.5 | 16.5 | 7.1 | 28.9 | 4.4 | 15.7 | 7.6 | 28.5 |
| 3–5 | 0.5 | 315 | 135 | 1350 | 225 | 8.1 | 4.1 | 13.5 | 8.4 | 37 | 4.5 | 14.6 | 7.8 | 34.2 |
| 3–6 | 0.5 | 315 | 135 | 1350 | 225 | 9.5 | 3.4 | 11.7 | 8.2 | 34 | 3.6 | 12.5 | 8.0 | 31.5 |
| 3–7 | 0.5 | 315 | 135 | 1350 | 225 | 10.8 | 2.9 | 9.5 | 7.5 | 28.6 | 2.7 | 10.4 | 8.1 | 27.4 |

## 3.5. SEM

The best amount of NaOH-activated AAMs (5%NaOH) were selected as their RFA activators for SEM analysis based on previous experiments [25]; surface images of the mixture are shown in Fig 8. Fig 8C depict the interface (B) between aggregate (A) and gel (C) in the alkali activated mortar. In the presence of 10% F and 5% NaOH, the geopolymer gel completely envelops the aggregate without any distinct boundaries (see Fig 8c). A chemical reaction may occur between the aggregate and the alkaline environment, or a mechanical mesh may take place between the two phases. The robust chemical bond between the geopolymer gel and aggregate could be the underlying reason for the high flexural and compressive strength observed in RFA mortar. A similar coating of aggregates was observed in 10%F microcracks (Fig 8a); this could be attributed to the elevated temperature during the solidification process.

A typical composite material made from fly ash consists primarily of aluminosilicate gel or geopolymer paste, along with unreacted fly ash particles and voids. The densest microstructure was observed in samples with visible aluminosilicate gels (Fig 8a, 8d, point 1, 3). Despite its high reactivity, the gel had an irregular shape with unreacted spherical fly ash particles present. Gel with a certain amount of unreacted or partially reacted particles in the mortar (Fig 8b, 8d, points 2, 4) exhibited a coarser microstructure and higher porosity, which indicates a moderate level of geopolymer reactivity in the sample.

Fewer floc-like products were observed in 10%F than in 10%F-5%NaOH (Fig 8a, 8c) under magnification 1.00 X, and the floc-like products mixed with sodium hydroxide bound into clumps more easily. At magnification of 2.00 X, the floc-like products on the surface of fly ash with sodium hydroxide activator were bound to each other, with larger particle size and

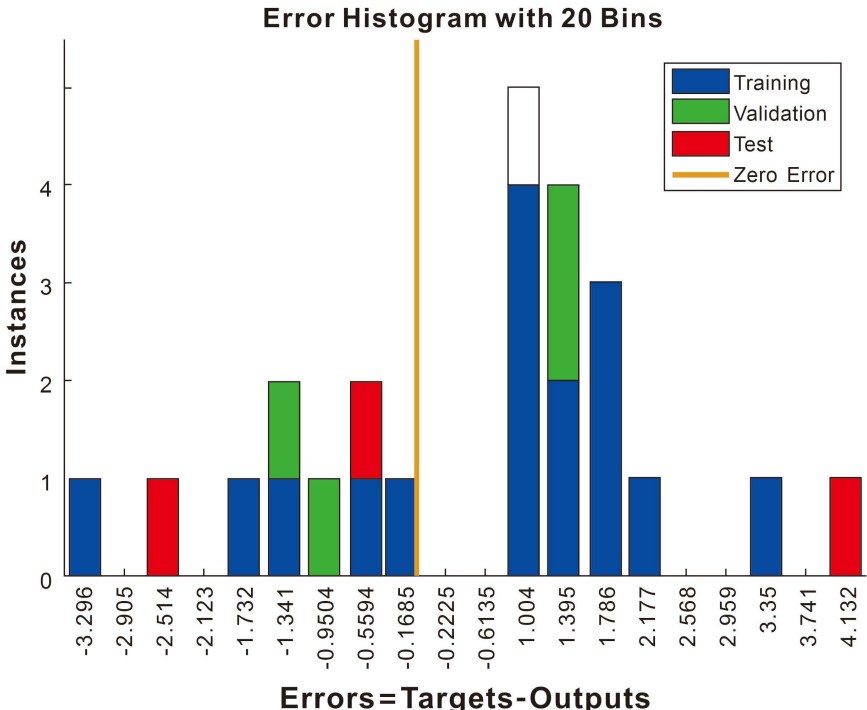

**Fig 7. Error histogram of neural network compressive strength prediction.**

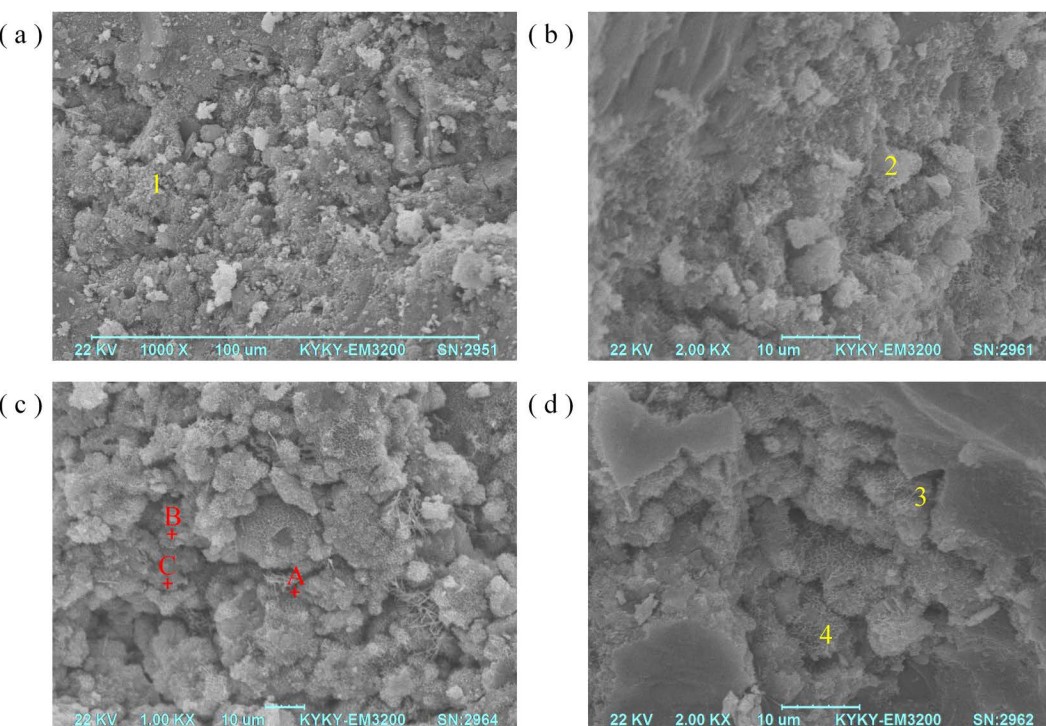

**Fig 8. SEM picture of fly ash-cement mortar at three days of treatment: (a,b) No alkali (10%F), (c,d) NaOH activated (10%F-5%NaOH).** A: Aggregate, B: Interface between gel and aggregate, C: Aluminosilicate/geopolymer paste, D: Microcracks.

smaller voids (Fig 8b and 8d). The 10%F-5%NaOH formed a visually more homogeneous gel structure, characterized by a denser and less porous matrix, and fewer partially reactive particles than 10%F.

## 3.6. Results of the correlation analysis

The correlation coefficients were greater than 0.5 except for the absolute values of fly ash and cement, and flexural strength and sodium hydroxide content (Fig 9), indicating that the flexural strength and compressive strength were moderately well correlated with the content of cement and other raw materials. Strength was negatively correlated with sodium hydroxide and fly ash, and positively correlated with cement content. NaoH was positively correlated with fly ash, but negatively correlated with the other parameters. Cement and fly ash was negatively correlated with NaOH, other are positive correlation.

## 3.7. Discussion

**Effect of sodium hydroxide content on fly ash cement strength.** The compressive and flexural strengths of alkali-activated mortars were significantly influenced by NaOH content (Table 2, Figs 3–4). At 3 days, mortars with 5% NaOH and 10% fly ash achieved the highest compressive (22.8 MPa) and flexural (5.4 MPa) strengths, surpassing the control group by 29% and 34%, respectively. This enhancement aligns with findings by Zhang et al. (2021) [24], who noted that alkali activators accelerate early hydration by disintegrating the vitreous structure of fly ash, releasing reactive silicate and aluminate ions. These ions rapidly form calcium silicate hydrate (C-S-H) gels, creating a dense three-dimensional network

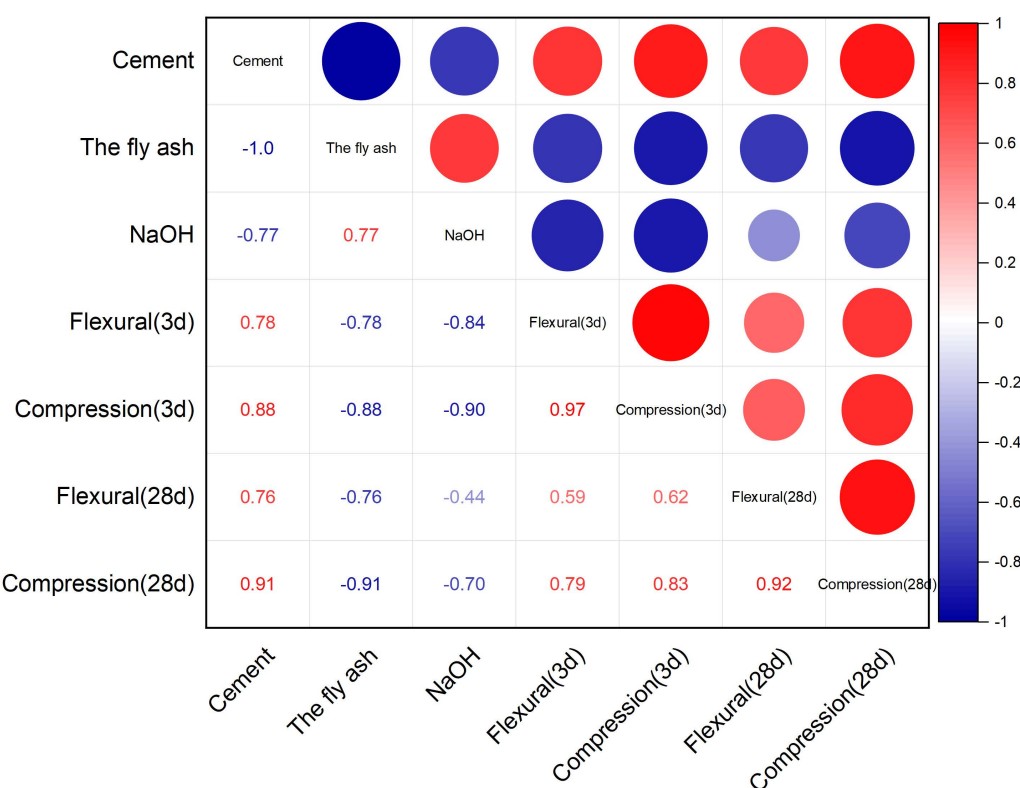

**Fig 9. Correlation analysis result chart.**

that improves early strength. However, beyond 5% NaOH, strength declined due to excessive alkalinity disrupting gel formation, a phenomenon also observed in Chen et al. (2018) [2].

By 28 days, optimal performance shifted to 6% NaOH and 10% fly ash, yielding compressive strength of 47.2 MPa. The slower late-stage strength gain (only 5% increase from 3 to 28 days) suggests that NaOH primarily accelerates early reactions, while later strength relies on prolonged pozzolanic activity. Notably, mortars with 30% fly ash exhibited the largest relative strength gain (37 MPa at 28 days), indicating that higher fly ash content enhances long-term performance despite initial slower reactivity.

**Age-dependent strength development.**  The divergent strength trends between 3 and 28 days (Figs 4–6) highlight the dual role of NaOH: it accelerates early gel formation but may inhibit late-stage hydration due to residual alkalinity. This aligns with [10], who reported that excess NaOH can lead to efflorescence, weakening interfacial bonds. SEM analysis (Fig 8) corroborated this: mortars with 5% NaOH displayed denser microstructures with fewer voids and unreacted particles compared to non-alkali samples, where loose, porous structures dominated.

**Neural network prediction and correlation analysis.**  The ANN model predicted compressive strength with ≤10% error (Fig 7), demonstrating its utility in optimizing mix designs. Input parameters (cement, fly ash, sand, water, NaOH) showed strong correlations with strength (Fig 9), except for NaOH, which exhibited a negative correlation at higher concentrations. This reflects the non-linear relationship between alkali dosage and performance, consistent with [25]. The model's accuracy supports its adoption in industrial settings to reduce trial-and-error experiments.

**Microstructural analysis via SEM.**  SEM images (Fig 8) revealed that NaOH-activated mortars formed continuous aluminosilicate gels enveloping aggregates, whereas non-alkali samples showed disjointed interfaces and microcracks. The homogeneous gel structure in 10% fly ash–5% NaOH mixtures explains their superior mechanical performance, as dense matrices reduce stress concentration points. These findings mirror [43], who linked microstructural compactness to durability in alkali-activated systems.

**Economic and environmental implications.**  Alkali-activated fly ash cement costs 200 RMB/t, 30–40% lower than Portland cement [10]. Using waste-derived NaOH further reduces expenses, while diverting fly ash from landfills mitigates environmental pollution. However, variability in fly ash composition (e.g., CaO content, unburned carbon) affects consistency, necessitating quality control protocols for industrial adoption.

## 4.  Conclusions

Our results showed that adding an alkaline activator into fly ash cement system can increase concrete's early activity and strength, which could have significant economic benefits. An ANN model can predict the strength of fly ash concrete mixed with an alkali activator with minor errors, which can reduce costs.

Optimal Mix Proportions: NaOH (5–6%) and fly ash (10–30%) synergistically enhance early and late-stage strength, with 10% fly ash–5% NaOH yielding peak 3-day performance and 10% fly ash–6% NaOH optimal at 28 days.

Mechanistic Insights: NaOH accelerates early C-S-H gel formation but may hinder long-term hydration. Microstructural homogeneity directly correlates with mechanical strength.

ANN Utility: The neural network model reliably predicts strength (≤10% error), offering a cost-effective tool for mix optimization.

Sustainability: Alkali-activated fly ash systems reduce costs and $CO_2$ emissions but require standardized fly ash quality for scalability.

## 5.  Recommendations and future perspectives

### 5.1.  Recommendations

Optimal Mix Design Implementation: Based on the findings, industrial applications should consider utilizing 5–6% sodium hydroxide (NaOH) as an alkali activator alongside 10–30% fly ash (depending on the desired curing period) to optimize

early and late-stage concrete strength. This combination balances economic and environmental benefits while enhancing mechanical performance.

Neural Network Integration: The developed artificial neural network (ANN) model, with a prediction error of ≤10%, should be adopted in preliminary mix design stages to reduce experimental workloads and accelerate material optimization. Further refinement of the model by incorporating additional variables (e.g., curing temperature, particle size distribution) could enhance its applicability.

### 5.2. Future perspectives

Exploration of Hybrid Alkali Activators: Future studies should investigate synergistic effects of combining NaOH with sodium silicate or other alkaline solutions to improve mechanical properties and durability, particularly in high-calcium fly ash systems.

Extended Curing and Durability Testing: Long-term performance (e.g., 90–365 days) and durability under aggressive environments (e.g., sulfate exposure, freeze-thaw cycles) should be evaluated to assess the feasibility of AAMs in infrastructure projects.

By addressing these gaps, future research can advance the industrial adoption of alkali-activated fly ash concrete as a sustainable alternative to conventional cement, aligning with global decarbonization goals.

## Acknowledgments

We are grateful to the teachers and students who participated in the field collection of data.

## Author contributions

**Conceptualization:** Fei Peng Liu, Shu Cheng Tan.

**Data curation:** Jing Xu, Fei Peng Liu.

**Investigation:** Jian Xin Zhao.

**Methodology:** Ai Min Gong.

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
