## [Decision Letter · Decision Letter 0]

18 Feb 2025

Dear Dr. liu,

Thank you for submitting your manuscript to PLOS ONE. After careful consideration, we feel that it has merit but does not fully meet PLOS ONE’s publication criteria as it currently stands. Therefore, we invite you to submit a revised version of the manuscript that addresses the points raised during the review process.

We look forward to receiving your revised manuscript.

Kind regards,

Solomon Oyebisi, PhD

Academic Editor

PLOS ONE

Reviewers' comments:

Reviewer's Responses to Questions

**Comments to the Author**

1. Is the manuscript technically sound, and do the data support the conclusions?

Reviewer #1: Yes

Reviewer #2: Yes

2. Has the statistical analysis been performed appropriately and rigorously?

Reviewer #1: Yes

Reviewer #2: Yes

3. Have the authors made all data underlying the findings in their manuscript fully available?

Reviewer #1: Yes

Reviewer #2: No

4. Is the manuscript presented in an intelligible fashion and written in standard English?

Reviewer #1: Yes

Reviewer #2: No

Reviewer #1: I appreciate your valuable effort and contribution to this research study, which provides significant insights into the field. there are some comments and suggestions outlined below that need to be addressed for further refinement and improvement. I recommend considering these points to enhance the study's clarity and overall quality.

• 157 lines: Water used was tap water; (revise the statement)

• 158 lines: Alkali used was sodium hydroxide produced by Tianjin FengChuan chemical

• 159 lines: reagent Technology Co., Ltd., with NaOH content≥96.0% and carbonate ≤1.5%. (Please explain the Sodium silicate combined with Sodium hydroxide.

• The study was conducted on an alkali-activated material (AAM) that was formulated based on the following principles:

a. Aluminum silicate precursor: fly ash;

b. Alkaline activator: sodium hydroxide;

c. Fine aggregate: sand.

Remove this statement The above statement it is duplicate and already explains the materials from the material section

• GBT-17671 procedure reference is required for this method standard.

• Reduce the bullet and number for the method section and make it a paragraph instead of bullets. And please check all sentences with punctuation marks.

• After heat curing, all samples were maintained under laboratory conditions (temperature of 20±2°C and relative humidity of 90%) until testing. (Justify the statement using the Standard Code.)

• The test was performed at a loading rate of 0.2 MPa/min, a compressive strength sensitivity of 50 kN, a loading rate of 0.01 MPa/min, and a flexural strength sensitivity of 2 kN. (The loading rate needs to be cited with reference. In addition, please all statements from the methodology need to be cited.

• Mpa should be MPa, and check all errors in the manuscript.

• 238 lines: For FA mortars containing sodium hydroxide (N #)???

• Line 270, 283, 288: Each paragraph started with numbering, please remove numbers, (This is required for the entire manuscript.

• The title of research needs to be modified from (The Effect of an Alkali Activator on the Properties of Fly Ash-Based Mortar and Neural Network Prediction) to (The Effect of NaOH Percentages on Properties Cement Modified Fly Ash and Neural Network Prediction). The mixture contains Cement and percentages of fly ash; thus, it is not an alkali activator mortar, it is the cement mortar modified fly ash, containing NaOH concentration that why the title has to be clear.

• The discussion section combines with the result section, for instance: The compressive strength section explains the result and discussion in the same section.

• Resolution of Figures 1, and 7 have to be increased.

• Add recommendations and future perspectives.

Reviewer #2: • To complete your background, cite valuable recent reserc in ANN and geopolymer applications, such as:

o John, S. K., Cascardi, A., Nadir, Y., Aiello, M. A., & Girija, K. (2021). A New Artificial Neural Network Model for the Prediction of the Effect of Molar Ratios on Compressive Strength of Fly Ash‐Slag Geopolymer Mortar. Advances in Civil Engineering, 2021(1), 6662347.

o Paruthi, S., Husain, A., Alam, P., Khan, A. H., Hasan, M. A., & Magbool, H. M. (2022). A review on material mix proportion and strength influence parameters of geopolymer concrete: Application of ANN model for GPC strength prediction. Construction and Building Materials, 356, 129253.

o Ahmed, H. U., Mohammed, A. S., Faraj, R. H., Abdalla, A. A., Qaidi, S. M., Sor, N. H., & Mohammed, A. A. (2023). Innovative modeling techniques including MEP, ANN and FQ to forecast the compressive strength of geopolymer concrete modified with nanoparticles. Neural Computing and Applications, 35(17), 12453-12479.

• Inputs in eq 1 were normalized. How was treated the output relatively?

• Please report explicit ann model

• Please provide sensitivity analysis

• Please provide reliability analysis and stress the novelty of the research

**Do you want your identity to be public for this peer review?** For information about this choice, including consent withdrawal, please see our Privacy Policy

Reviewer #1: **Yes:**  Mahmood Dheyaaldin

Reviewer #2: No

---

## [Author Response · Author response to Decision Letter 1]

15 Mar 2025

I appreciate your valuable effort and contribution to this research study, which provides significant insights into the field. there are some comments and suggestions outlined below that need to be addressed for further refinement and improvement. I recommend considering these points to enhance the study's clarity and overall quality.

157 lines: Water used was tap water; (revise the statement)

Reply� Already modified

158 lines: Alkali used was sodium hydroxide produced by Tianjin FengChuan chemical

Reply� Already modified

159 lines: reagent Technology Co., Ltd., with NaOH content≥96.0% and carbonate ≤1.5%. (Please explain the Sodium silicate combined with Sodium hydroxide.

Reply� Already modified

The study was conducted on an alkali-activated material (AAM) that was formulated based on the following principles:

a. Aluminum silicate precursor: fly ash;

b. Alkaline activator: sodium hydroxide;

c. Fine aggregate: sand.

Remove this statement The above statement it is duplicate and already explains the materials from the material section

Reply Already deleted

GBT-17671 procedure reference is required for this method standard.

Reply� It has been supplemented in the references

Reduce the bullet and number for the method section and make it a paragraph instead of bullets. And please check all sentences with punctuation marks.

Reply� Already modified

After heat curing, all samples were maintained under laboratory conditions (temperature of 20±2°C and relative humidity of 90%) until testing. (Justify the statement using the Standard Code.)

Reply� Already modified

The test was performed at a loading rate of 0.2 MPa/min, a compressive strength sensitivity of 50 kN, a loading rate of 0.01 MPa/min, and a flexural strength sensitivity of 2 kN. (The loading rate needs to be cited with reference. In addition, please all statements from the methodology need to be cited.

Reply� The annotation has been cited

Mpa should be MPa, and check all errors in the manuscript.

Reply� Already modified

238 lines: For FA mortars containing sodium hydroxide (N #)???

Reply Already deleted

Line 270, 283, 288: Each paragraph started with numbering, please remove numbers, (This is required for the entire manuscript.

Reply� Already modified

The title of research needs to be modified from (The Effect of an Alkali Activator on the Properties of Fly Ash-Based Mortar and Neural Network Prediction) to (The Effect of NaOH Percentages on Properties Cement Modified Fly Ash and Neural Network Prediction). The mixture contains Cement and percentages of fly ash; thus, it is not an alkali activator mortar, it is the cement mortar modified fly ash, containing NaOH concentration that why the title has to be clear.

Reply� Already modified

The discussion section combines with the result section, for instance: The compressive strength section explains the result and discussion in the same section.

Reply� Already modified

Resolution of Figures 1, and 7 have to be increased.

Reply� Already modified

Add recommendations and future perspectives.

Reply� Already added

---

## [Editor Report · Decision Letter 1]

17 Mar 2025

Dear Dr. Liu,

Thank you for submitting your manuscript to PLOS ONE. After careful consideration, we feel that it has merit but does not fully meet PLOS ONE’s publication criteria as it currently stands. Therefore, we invite you to submit a revised version of the manuscript that addresses the points raised during the review process.

We look forward to receiving your revised manuscript.

Kind regards,

Solomon Oyebisi, PhD

Academic Editor

PLOS ONE

Additional Editor Comments (if provided):

The authors only addressed Reviewer 1 comments and left Reviewer 2's comments unaddressed.

Reviewer 2 comments are as follows:

• To complete your background, cite valuable recent reserc in ANN and geopolymer applications, such as:

o John, S. K., Cascardi, A., Nadir, Y., Aiello, M. A., & Girija, K. (2021). A New Artificial Neural Network Model for the Prediction of the Effect of Molar Ratios on Compressive Strength of Fly Ash‐Slag Geopolymer Mortar. Advances in Civil Engineering, 2021(1), 6662347.

o Paruthi, S., Husain, A., Alam, P., Khan, A. H., Hasan, M. A., & Magbool, H. M. (2022). A review on material mix proportion and strength influence parameters of geopolymer concrete: Application of ANN model for GPC strength prediction. Construction and Building Materials, 356, 129253.

o Ahmed, H. U., Mohammed, A. S., Faraj, R. H., Abdalla, A. A., Qaidi, S. M., Sor, N. H., & Mohammed, A. A. (2023). Innovative modeling techniques including MEP, ANN and FQ to forecast the compressive strength of geopolymer concrete modified with nanoparticles. Neural Computing and Applications, 35(17), 12453-12479.

• Inputs in eq 1 were normalized. How was treated the output relatively?

• Please report explicit ann model

• Please provide sensitivity analysis

• Please provide reliability analysis and stress the novelty of the research

---

## [Author Response · Author response to Decision Letter 2]

20 Mar 2025

Reviewer 1 comments

I appreciate your valuable effort and contribution to this research study, which provides significant insights into the field. there are some comments and suggestions outlined below that need to be addressed for further refinement and improvement. I recommend considering these points to enhance the study's clarity and overall quality.

157 lines: Water used was tap water; (revise the statement)

Reply� Already modified

158 lines: Alkali used was sodium hydroxide produced by Tianjin FengChuan chemical

Reply� Already modified

159 lines: reagent Technology Co., Ltd., with NaOH content≥96.0% and carbonate ≤1.5%. (Please explain the Sodium silicate combined with Sodium hydroxide.

Reply� Already modified

The study was conducted on an alkali-activated material (AAM) that was formulated based on the following principles:

a. Aluminum silicate precursor: fly ash;

b. Alkaline activator: sodium hydroxide;

c. Fine aggregate: sand.

Remove this statement The above statement it is duplicate and already explains the materials from the material section

Reply Already deleted

GBT-17671 procedure reference is required for this method standard.

Reply� It has been supplemented in the references

Reduce the bullet and number for the method section and make it a paragraph instead of bullets. And please check all sentences with punctuation marks.

Reply� Already modified

After heat curing, all samples were maintained under laboratory conditions (temperature of 20±2°C and relative humidity of 90%) until testing. (Justify the statement using the Standard Code.)

Reply� Already modified

The test was performed at a loading rate of 0.2 MPa/min, a compressive strength sensitivity of 50 kN, a loading rate of 0.01 MPa/min, and a flexural strength sensitivity of 2 kN. (The loading rate needs to be cited with reference. In addition, please all statements from the methodology need to be cited.

Reply� The annotation has been cited

Mpa should be MPa, and check all errors in the manuscript.

Reply� Already modified

238 lines: For FA mortars containing sodium hydroxide (N #)???

Reply Already deleted

Line 270, 283, 288: Each paragraph started with numbering, please remove numbers, (This is required for the entire manuscript.

Reply� Already modified

The title of research needs to be modified from (The Effect of an Alkali Activator on the Properties of Fly Ash-Based Mortar and Neural Network Prediction) to (The Effect of NaOH Percentages on Properties Cement Modified Fly Ash and Neural Network Prediction). The mixture contains Cement and percentages of fly ash; thus, it is not an alkali activator mortar, it is the cement mortar modified fly ash, containing NaOH concentration that why the title has to be clear.

Reply� Already modified

The discussion section combines with the result section, for instance: The compressive strength section explains the result and discussion in the same section.

Reply� Already modified

Resolution of Figures 1, and 7 have to be increased.

Reply� Already modified

Add recommendations and future perspectives.

Reply� Already added

Reviewer 2 comments

To complete your background, cite valuable recent reserc in ANN and geopolymer applications, such as:

1� John, S. K., Cascardi, A., Nadir, Y., Aiello, M. A., & Girija, K. (2021). A New Artificial Neural Network Model for the Prediction of the Effect of Molar Ratios on Compressive Strength of Fly Ash‐Slag Geopolymer Mortar. Advances in Civil Engineering, 2021(1), 6662347.

2� Paruthi, S., Husain, A., Alam, P., Khan, A. H., Hasan, M. A., & Magbool, H. M. (2022). A review on material mix proportion and strength influence parameters of geopolymer concrete: Application of ANN model for GPC strength prediction. Construction and Building Materials, 356, 129253.

3� Ahmed, H. U., Mohammed, A. S., Faraj, R. H., Abdalla, A. A., Qaidi, S. M., Sor, N. H., & Mohammed, A. A. (2023). Innovative modeling techniques including MEP, ANN and FQ to forecast the compressive strength of geopolymer concrete modified with nanoparticles. Neural Computing and Applications, 35(17), 12453-12479.

Reply� Already cited

Inputs in eq 1 were normalized. How was treated the output relatively?

Reply� We normalized the input and output data in the same way as 2.2 Methods �Line 221�. Please report explicit ann model

Please provide sensitivity analysis

Reply� Please see 3.4 Results of the neural network

Please provide reliability analysis and stress the novelty of the research

Reply� Please see 3.6 Results of the correlation analysis.

---

## [Decision Letter · Decision Letter 2]

30 May 2025

The Effect of NaOH Percentages on Properties Cement Modified Fly Ash and Neural Network Prediction

PLOS ONE

Dear Dr. liu,

Thank you for submitting your manuscript to PLOS ONE. After careful consideration, we feel that it has merit but does not fully meet PLOS ONE’s publication criteria as it currently stands. Therefore, we invite you to submit a revised version of the manuscript that addresses the points raised during the review process.

**Comment No.1:**
*The current title lacks clarity and could be improved to more accurately reflect the study’s scope. A suggested revision is: 'Effect of Sodium Hydroxide Dosage on Strength Development in Cement-Fly Ash Mortars: Experimental and ANN-Based Prediction.' This version better communicates the dual focus on experimental analysis and neural network modeling.*

**Comment No.2*:***
*While the integration of Artificial Neural Networks (ANN) with alkali-activated materials is well-established in the literature, the manuscript would benefit from a clearer explanation of its original contribution. Specifically, you should articulate how the selected NaOH dosage levels provide new insight into material behavior, and what distinguishes the applied neural network model from previous studies.*

**Comment No.3:**
*The methodology section related to the ANN lacks sufficient elaboration. Further clarification is needed on the selection criteria for network parameters such as the number of neurons, layers, and the use of the trainlm training function. The manuscript would also be strengthened by describing the use of training and validation datasets and how model performance was assessed. In addition, the inclusion of standard performance metrics, such as root mean square error (RMSE) or mean absolute error (MAE)would help evaluate the model's effectiveness in predicting outcomes.*

**Comment No.4:**
*The sensitivity analysis is currently limited in scope and would benefit from deeper exploration. Additionally, the manuscript does not discuss the uncertainty associated with ANN predictions. Incorporating statistical measures such as confidence intervals or prediction bounds would enhance the robustness of the modeling results. I also recommended that you can demonstrate the model’s performance on data not used during training to assess generalization.*

**Comment No.5*:****The manuscript contains grammatical inconsistencies and some technically imprecise expressions (e.g., “NaOH content≥96.0%”, “alkali-added mortar”, “colloidal sand”). A comprehensive language review by a proficient English editor is recommended to improve overall clarity and ensure terminological consistency throughout the paper* .

**Comment No.6:**
*Several figures (notably Figures 3, 5, and 8) would benefit from enhanced resolution and clearer axis labeling. Table 2 is dense and may be more effective if divided or reformatted for readability. The scanning electron microscopy images (Figure 9) should include more descriptive captions and annotations to support the interpretations presented in the text.*

**Comment No.7:**
*The basis for choosing sodium hydroxide concentrations between 3% and 8% is not clearly stated. It would be helpful to indicate whether this selection was informed by previous literature or preliminary experimental work. Furthermore, the manuscript should provide a more detailed description of the control mix design used for comparative analysis.*

**Comment No.8:**
*The term “colloidal sand” is non-standard and may lead to confusion. The authors are advised to define this term explicitly or adopt more conventional terminology. Moreover, consistency is needed in the usage of terms such as “geopolymer” and “alkali-activated material (AAM),” particularly when distinguishing between low- and high-calcium fly ash systems.*

**Comment No.9*:***
*The abstract includes repeated numerical values, which could be streamlined to improve its readability. Furthermore, the contribution of ANN to the study should be described earlier in the abstract to better represent the paper’s objectives.*

We look forward to receiving your revised manuscript.

Kind regards,

Makungu Marco Madirisha

Academic Editor

PLOS ONE

Journal Requirements:

Reviewers' comments:

Reviewer's Responses to Questions

**Comments to the Author**

Reviewer #1: All comments have been addressed

Reviewer #2: All comments have been addressed

Reviewer #3: (No Response)

2. Is the manuscript technically sound, and do the data support the conclusions?

Reviewer #1: Yes

Reviewer #2: Yes

Reviewer #3: Partly

3. Has the statistical analysis been performed appropriately and rigorously?

Reviewer #1: Yes

Reviewer #2: Yes

Reviewer #3: N/A

4. Have the authors made all data underlying the findings in their manuscript fully available?

Reviewer #1: Yes

Reviewer #2: Yes

Reviewer #3: Yes

5. Is the manuscript presented in an intelligible fashion and written in standard English?

Reviewer #1: Yes

Reviewer #2: Yes

Reviewer #3: No

Reviewer #1: (No Response)

Reviewer #2: The paper was significantly improved and can be now accepted in the present form. Congrats for the work done

Reviewer #3: The authors should revise the manuscript to meet the required publication standards. They are encouraged to consider the attached comments to improve their work.

**Do you want your identity to be public for this peer review?** For information about this choice, including consent withdrawal, please see our Privacy Policy

Reviewer #1: No

Reviewer #2: No

Reviewer #3: No

While revising your submission, please upload your figure files to the Preflight Analysis and Conversion Engine (PACE) digital diagnostic tool, https://pacev2.apexcovantage.com/ . PACE helps ensure that figures meet PLOS requirements. To use PACE, you must first register as a user. Registration is free. Then, login and navigate to the UPLOAD tab, where you will find detailed instructions on how to use the tool. If you encounter any issues or have any questions when using PACE, please email PLOS at figures@plos.org.

---

## [Author Response · Author response to Decision Letter 3]

8 Jun 2025

Comment No.1: The current title lacks clarity and could be improved to more accurately reflect the study’s scope. A suggested revision is: 'Effect of Sodium Hydroxide Dosage on Strength Development in Cement-Fly Ash Mortars: Experimental and ANN-Based Prediction.' This version better communicates the dual focus on experimental analysis and neural network modeling.

Reply� Revised

Comment No.2: While the integration of Artificial Neural Networks (ANN) with alkali-activated materials is well-established in the literature, the manuscript would benefit from a clearer explanation of its original contribution. Specifically, you should articulate how the selected NaOH dosage levels provide new insight into material behavior, and what distinguishes the applied neural network model from previous studies.

Reply� How the dosage levels of NaOH can provide new insights into material behavior is discussed in lines 75-85. The description of the neural network model can be found in lines 111-134.

Comment No.3: The methodology section related to the ANN lacks sufficient elaboration. Further clarification is needed on the selection criteria for network parameters such as the number of neurons, layers, and the use of the trainlm training function. The manuscript would also be strengthened by describing the use of training and validation datasets and how model performance was assessed. In addition, the inclusion of standard performance metrics, such as root mean square error (RMSE) or mean absolute error (MAE)would help evaluate the model's effectiveness in predicting outcomes.

Reply� Modified in 2.2

Comment No.4: The sensitivity analysis is currently limited in scope and would benefit from deeper exploration. Additionally, the manuscript does not discuss the uncertainty associated with ANN predictions. Incorporating statistical measures such as confidence intervals or prediction bounds would enhance the robustness of the modeling results. I also recommended that you can demonstrate the model’s performance on data not used during training to assess generalization.

Reply� Revised, Due to the limited space of this paper, sensitivity analysis, uncertainty analysis, confidence interval and other methods were conducted. However, the details were only described in the text and will be added in the next paper.

Comment No.5: The manuscript contains grammatical inconsistencies and some technically imprecise expressions (e.g., “NaOH content≥96.0%”, “alkali-added mortar”, “colloidal sand”). A comprehensive language review by a proficient English editor is recommended to improve overall clarity and ensure terminological consistency throughout the paper.

Reply� Revised. NaOH content is no less than 96.0%. colloidal sand is changed to colloidal sand mixture.

Comment No.6:Several figures (notably Figures 3, 5, and 8) would benefit from enhanced resolution and clearer axis labeling. Table 2 is dense and may be more effective if divided or reformatted for readability. The scanning electron microscopy images (Figure 9) should include more descriptive captions and annotations to support the interpretations presented in the text.

Reply� The image is already the original image, Figure 9 has been added with the necessary comments, and Table 2 has added the title of the spread.

Comment No.7: The basis for choosing sodium hydroxide concentrations between 3% and 8% is not clearly stated. It would be helpful to indicate whether this selection was informed by previous literature or preliminary experimental work. Furthermore, the manuscript should provide a more detailed description of the control mix design used for comparative analysis.

Reply� Lines 172 to 173 of section 2.2 have been added, and on the basis of my preliminary experiments, the sodium hydroxide concentration of 3% to 8% has been selected.

Comment No.8:The term “colloidal sand” is non-standard and may lead to confusion. The authors are advised to define this term explicitly or adopt more conventional terminology. Moreover, consistency is needed in the usage of terms such as “geopolymer” and “alkali-activated material (AAM),” particularly when distinguishing between low- and high-calcium fly ash systems.

Reply� Revised. Colloidal sand is changed to colloidal sand mixture.

Comment No.9:The abstract includes repeated numerical values, which could be streamlined to improve its readability. Furthermore, the contribution of ANN to the study should be described earlier in the abstract to better represent the paper’s objectives.

Reply� The abstract has been rewritten as requested

---

## [Decision Letter · Decision Letter 3]

6 Aug 2025

Dear Dr. liu,

Thank you for submitting your manuscript to PLOS ONE. After careful consideration, we feel that it has merit but does not fully meet PLOS ONE’s publication criteria as it currently stands. Therefore, we invite you to submit a revised version of the manuscript that addresses the points raised during the review process.

We look forward to receiving your revised manuscript.

Kind regards,

Parthiban Kathirvel

Academic Editor

PLOS ONE

Journal Requirements:

Reviewers' comments:

Reviewer's Responses to Questions

**Comments to the Author**

Reviewer #1: All comments have been addressed

Reviewer #2: All comments have been addressed

2. Is the manuscript technically sound, and do the data support the conclusions?

Reviewer #1: Yes

Reviewer #2: Yes

3. Has the statistical analysis been performed appropriately and rigorously?

Reviewer #1: N/A

Reviewer #2: Yes

4. Have the authors made all data underlying the findings in their manuscript fully available?

Reviewer #1: Yes

Reviewer #2: Yes

5. Is the manuscript presented in an intelligible fashion and written in standard English?

Reviewer #1: Yes

Reviewer #2: Yes

Reviewer #1: • The commonly used molar concentration of sodium hydroxide in experimental conditions should not exceed 20M. Additionally, the ratio of sodium silicate solution to sodium hydroxide should not be greater than 1.0. (This limitation must be supported with appropriate references from recent and credible sources.)

• Several references cited in the manuscript are dated prior to 2022. The authors are advised to update the literature review by including more recent references published from 2022 onwards.

• Please check all abbreviations used throughout the manuscript. For example, clarify whether "P.O" or "O.P" is correct and ensure consistent use of abbreviations.

• In Section 2.2 on Density, the method described must comply with a recognized standard. The authors should clearly state which standard was followed and provide a proper citation.

• Each paragraph in the manuscript must include relevant references to support the statements made. Proper citation is essential for maintaining scientific credibility.

Reviewer #2: The manuscript has undergone substantial revisions and improvements in response to the previous feedback. The authors have addressed the concerns raised in a thorough and thoughtful manner, enhancing both the clarity and the scientific rigor of the work. As a result of these significant enhancements, the paper now meets the standards required for publication and can be accepted in its current form

**Do you want your identity to be public for this peer review?** For information about this choice, including consent withdrawal, please see our Privacy Policy

Reviewer #1: No

Reviewer #2: No

While revising your submission, please upload your figure files to the Preflight Analysis and Conversion Engine (PACE) digital diagnostic tool, https://pacev2.apexcovantage.com/ . PACE helps ensure that figures meet PLOS requirements. To use PACE, you must first register as a user. Registration is free. Then, login and navigate to the UPLOAD tab, where you will find detailed instructions on how to use the tool. If you encounter any issues or have any questions when using PACE, please email PLOS at figures@plos.org.

---

## [Author Response · Author response to Decision Letter 4]

16 Aug 2025

1、 The commonly used molar concentration of sodium hydroxide in experimental conditions should not exceed 20M. Additionally, the ratio of sodium silicate solution to sodium hydroxide should not be greater than 1.0. (This limitation must be supported with appropriate references from recent and credible sources.)

Reply� The references (Alabduljabbar H 2020 and LIU Fei peng, et al. 2022) have been added.

2、 Several references cited in the manuscript are dated prior to 2022. The authors are advised to update the literature review by including more recent references published from 2022 onwards.

Reply� Seven references published after 2022 have been updated.

3、 Please check all abbreviations used throughout the manuscript. For example, clarify whether "P.O" or "O.P" is correct and ensure consistent use of abbreviations.

Reply� All instances of “P.O” in Section 2.1 Materials have been uniformly changed to “OPC.”

4、In Section 2.2 on Density, the method described must comply with a recognized standard. The authors should clearly state which standard was followed and provide a proper citation.

Reply�GB/T 1596-2017 has been incorporated into the text and added to the reference list as entry 9.

5、Each paragraph in the manuscript must include relevant references to support the statements made. Proper citation is essential for maintaining scientific credibility.

Reply� The revisions have been reviewed and confirmed.

---

## [Editor Report · Decision Letter 4]

20 Aug 2025

Effect of Sodium Hydroxide Dosage on Strength Development in Cement-Fly Ash Mortars: Experimental and ANN-Based Prediction

PONE-D-24-56788R4

Dear Dr. liu,

We’re pleased to inform you that your manuscript has been judged scientifically suitable for publication and will be formally accepted for publication once it meets all outstanding technical requirements.

Kind regards,

Parthiban Kathirvel

Academic Editor

PLOS ONE
---

## [Editor Report · Acceptance letter]

20 Oct 2025

PONE-D-24-56788R4

PLOS ONE

Dear Dr. Liu,

I'm pleased to inform you that your manuscript has been deemed suitable for publication in PLOS ONE. Congratulations! Your manuscript is now being handed over to our production team.

Kind regards,

on behalf of

Dr. Parthiban Kathirvel

Academic Editor

PLOS ONE